# Elements of Effective Population Surveillance Systems for Monitoring Obesity in School Aged Children

**DOI:** 10.3390/ijerph17186812

**Published:** 2020-09-18

**Authors:** Louise L. Hardy, Seema Mihrshahi

**Affiliations:** 1Faculty of Medicine and Health, The University of Sydney, Sydney, NSW 2006, Australia; 2Department of Health Systems and Populations, Faculty of Medicine, Health and Human Sciences, Macquarie University, Sydney, NSW 2190, Australia; seema.mihrshahi@mq.edu.au

**Keywords:** child, adolescent, surveillance, nutrition, physical activity, sedentary behaviour, sleep

## Abstract

The continuing high prevalence of child overweight and obesity globally means that it remains the most common chronic health condition in children. Population-based child obesity surveillance systems are critical for monitoring trends in obesity and related behaviours, and determining the overall effect of child obesity prevention strategies. Effective surveillance systems may vary in methods, scope, purpose, objectives, and attributes, and our aim was to provide an overview of child obesity surveillance systems globally, and to highlight main components and other types of survey data that can enhance our understanding of child obesity. Measures of adiposity, including body mass index and waist circumference are essential, but effective surveillance must also include measures of weight-related behaviours, including diet, physical activity, sedentary time, and sleep. While objective measures are desirable, the variability in psychometrics and rapid evolution of wearable devices is potentially problematic for examining long-term trends over time and how behaviours may change. Questionnaires on self-reported behaviours are often used but also have limitations. Because the determinants of obesity are not only functioning at the individual level, some measures of the broader environmental and commercial determinants, including the built and food environments, are useful to guide upstream policy decisions.

## 1. Introduction

In the last quarter of the 20th century, an obesity pandemic emerged in many countries and is currently the most prevalent chronic health condition in children. In 1975 the global prevalence of obesity in children was ~0.8%, increasing to 5.6% in girls and 7.8% in boys in 2016 [1]. There are regional differences with higher prevalence among children from Polynesia, Micronesia, Central Latin America, Middle East and North Africa, and high-income English-speaking countries. The data show an emerging plateau in the prevalence among children from high-income regions but an increase in high-income Asian countries. There are also socioeconomic differences within jurisdictions; in high-income regions the prevalence is higher among children from low socioeconomic areas, conversely, in low- and middle-income countries the prevalence is higher children from high socioeconomic areas [2].

For many years there has been substantial investment by governments and public health practitioners to reduce the incidence of child obesity, but the prevalence of child obesity remains at concerning levels [1]. Obesity is a complex multifactorial condition arising from energy imbalance, which has led to extensive interventions that focus on improving weight-related behaviours. The most recent Cochrane meta-analysis of interventions to reduce childhood obesity shows there is only moderate certainty that interventions to improve diet and physical activity behaviours are effective [3]. While interventions are an important public health response, they are conducted among relatively small groups and fail to capture broader environmental and policy-related actions that may influence prevalence and incidence of child obesity at a population level.

In contrast, population child obesity surveillance systems are better equipped to examine the sum effect of all child obesity prevention investments. An obesity surveillance system comprises a series of surveys conducted at regular intervals, used to monitor long-term trends in obesity and overweight and related behaviours. It can be used to determine the prevalence rates among subgroups, track trends across time, and determine whether obesity prevention and management interventions are collectively effective. The data provide policy makers with the necessary epidemiological information to make evidence-based decisions.

Child obesity surveillance is conducted in many jurisdictions; however, these surveillance systems vary in methods, scope, purpose, objectives, and attributes, and what is considered important in one jurisdiction may be less important to another. While there are guidelines for evaluating public health surveillance systems (e.g., [4]) there is diversity in the data collected across child obesity surveillance. For example, some surveillance systems only collect body mass index (e.g., [5]) and others collect only some weight-related behaviours associated with the development of overweight and obesity. Table 1 lists a selective example showing the diversity in methods and measures of population health systems conducted in different countries that enable the surveillance and temporal monitoring of child obesity. An example case study of an Australian state-based child obesity monitoring survey that provided evidence to guide policy decisions is given in Box 1.

Box 1Case study of a successful serial state-based child obesity surveillance survey: The New South Wales Schools Physical Activity and Nutrition Survey.In 2002, in response to growing concerns on the perceived increase in child obesity, the New South Wales (NSW) Government convened a Child Obesity Summit [6]. One outcome was to establish a centre to determine and monitor the prevalence of weight status and indicators of their weight-related behaviours in a representative sample of NSW school children aged 5 to 16 years. The survey, the Schools Physical Activity and Nutrition Survey (SPANS) was conducted in 2004 [7], 2010 [8] and 2015 [9] in a random representative sample of schools and students using consistent methodology. The central purpose of SPANS has been to estimate the prevalence and trends in children’s weight status and weight-related behaviours to inform NSW State Health Plans and develop recommendations for the NSW Ministry of Health.SPANS was a school-based survey with trained field teams measuring children’s height, weight, and waist circumference, and measures of fitness and fundamental movement skills. Children’s health behaviours were collected via questionnaires (parent proxy report for children age <10 years), which included measures of diet and eating behaviour, physical activity, school travel, and more recently sleep and dental health. Participating schools completed a school environment questionnaire on the school nutrition and physical activity environment. Trends in overweight and obesity could be calculated from anthropometric data collected in 1985 (The Australian Health and Fitness Survey [10]) and in 1997 (NSW Schools Fitness and Physical Activity Survey [11]) with the same methodology as SPANS.The data collected throughout the SPANS series made a substantial contribution to the evidence-base for different policy areas and the development of programs. The trends in in overweight and obesity highlight that children from low socioeconomic status (SES) areas (especially adolescents) [12], cultural and linguistically diverse language backgrounds [13], and of Aboriginal and Torres Strait Island heritage [14] would benefit from targeted interventions or programs. In 2010, it was found that one in five children entering the first year of school had overweight and obesity [15], highlighting that weight-related behaviours are established in the child’s home environment. Statewide programs were then developed for the early childhood sector to help communicate to parents on healthy eating and physical activity [16]. The delivery of fundamental movement skills (FMS) programs in primary schools [17] to address children’s low proficiency in FMS [18,19] is included in the NSW school curriculum. The low prevalence of meeting physical activity recommends has been addressed through a universal voucher program (i.e., ActiveKids) to increase children’s participation in organised sports [20]. Fruit and vegetable breaks were implemented in primary schools to address children’s low consumption of these foods [17]. In 2015, SPANS showed that children who purchased food regularly from the school canteen were more likely to have overweight or obesity [21]. This led to a more rigorous implementation of the NSW Healthy School Canteen Strategy in 2017 [22].

Overweight and obesity are the outcome, so it is necessary to measure indicators of weight-related behaviours concurrently to determine what public health intervention programs are required. Ideally, surveillance surveys should include practices in the home environment associated with overweight/obesity (e.g., TV in the bedroom, soft drinks in the home, no screen time rules), which can be useful to encourage health-promoting parenting practices [23]. Because obesity-promoting behaviours are associated with other factors beyond the individual level, triangulating child obesity surveillance data with other data sources on the drivers of an obesogenic environment will provide a better understanding on developing environments to improve weight-related behaviours (e.g., urbanisation, transportation, food systems) [24]. The purpose of this paper is to describe the components and their rationale for inclusion in an effective child obesity surveillance system, and other types of survey data that can enhance our understanding of child obesity.

## 2. Considerations for Population Surveillance Surveys

Population health surveillance is the on-going systematic collection, analyses, interpretation, and, timely dissemination of the data to decision makers responsible for prevention and control [32]. Surveillance data can be used to measure the need for interventions and for directly measuring the effects of population-wide interventions. Active surveillance requires trained field teams to collect the data and while it provides the most accurate and timely information, active surveillance can be expensive. An alternative method to collect population-level information on child obesity is through passive surveillance and other health monitoring systems such as medical records. Passive surveillance is less expensive than active surveillance, but limited because data are not collected systematically and because the data are collected through a network of systems may have variability in the completeness, quality, and timeliness of the data.

Details on population surveillance survey designs and data analyses are beyond the scope of this paper, so we present a general overview of factors to consider. Table 1 shows there are different survey methods to monitor population-level child obesity, which include mobile health vans, households, and schools. The methodology differs across jurisdictions according to a range of factors, including expertise and funding. Surveys such as the US National Health and Nutrition Examination Survey [27] use mobile health vans staffed with health assessment specialists who collect extensive behavioural and biomarker data, but these are expensive to conduct. Household and school-based surveys are relatively low cost but are often unable to collect extensive behavioural and biomarker data.

The sampling frame must replicate the population for the estimates to be generalizable. For example, the sampling frame needs to ensure that all demographic characteristics of interest are covered, such as age, sex, socioeconomic status, geography (rural/urban), and cultural/racial background. A sample of children are randomly selected and invited to participate in the survey. Written consent is typically an ethics requirement; however, the governments in the United Kingdom [5] and the US state of Arkansas [33] have legislated data collection with opt-out consent for their childhood obesity monitoring and screening programs.

The measurement of weight-related behaviours and how to measure each indicator will differ across jurisdictions. The survey instruments need to reflect cultural traditions and customs of dietary and physical activity behaviours. One of the purposes of population surveillance is to examine temporal trends so the methodology must remain consistent across survey years. Ideally, data should be captured electronically using computers or hand-held and other mobile devices. In contrast to pen and paper, electronical data entry reduces errors through logic checks, missing data, and minimal cost (e.g., data entry–re-entry), and is ready for cleaning and analyses.

## 3. Adiposity

Gold standard measures of adiposity (e.g., DEXA, underwater weighing) are generally not feasible for most countries to use in population obesity surveillance. Similarly, skinfold thickness which measures subcutaneous body fat are difficult measurements to make with precision and accuracy without rigorous training, and there is no consensus on which sites should be used. Alternatively, height, weight, and waist measurements are widely accepted as simple proxy measures of body fat. Height (m) and weight (kg) are used to calculate body mass index (BMI: kg/m^2^) and are reasonably correlated with body fat in children [34].

In contrast to adult BMI values to define overweight (i.e., ≥25 kg/m^2^) and obesity (i.e., ≥30 kg/m^2^), child values are complicated by growth and development. There is currently no universal standard system for using BMI to define overweight and obesity in children. In a clinical setting, age- and sex-adjusted BMI z-scores standard deviations (SD) are used for screening and percentiles used to define overweight and obesity. The World Health Organization system is based on BMI z-score standard deviations (>2 SD for overweight and >3 SD for obesity) [35], the U.S. Center for Disease Control (CDC) and Prevention System is based on age- and sex-specific BMI percentiles (85th for overweight and 95th for obesity) [36]. For population surveillance, the International Obesity Task Force (IOTF) recommends using the international age- and sex-adjusted cut point values for BMI categories, which correspond to adult BMI values for determining the prevalence of overweight and obesity [37].

While each system gives somewhat different estimates of overweight and obesity prevalence (e.g., BMI z-score functions poorly as an indicator of adiposity among children with obesity [38]), the 85th and 95th percentile cut off points were arbitrarily selected and not based on evidence of adverse health outcomes. Additionally, BMI values for overweight and obesity differ across population groups, with lower BMI cut points for Asians [39] and higher BMI cut points for Polynesians [40] associated with morbidity.

The IOTF cut points for underweight, healthy weight, overweight, and obesity in children [37] have been recommended for international use to allow jurisdictional comparisons. The IOTF reference population used to determine cut points for BMI categories was developed on approximately 200,000 children from six countries in three continents, which potentially represent better growth patterns of international children compared with that of the CDC growth charts developed for US children.

While BMI is the most widely recognized surrogate of obesity, it does not provide information about the distribution of body fat. The distribution or patterning of body fat is a risk factor for chronic disease [41]. Central or abdominal accumulation of fat (i.e., android) is a greater risk factor for diabetes and heart disease than fat distribution in extremities (i.e., gynoid) [42]. Studies have shown that children’s waist circumferences have increased more than BMI over time [12,43,44], indicating the importance of monitoring abdominal obesity. The waist-to-height ratio (WHtR: cm/cm) is the best indicator of abdominal obesity with values ≥0.5 highly correlated with cardiometabolic diseases [45]. To better understand child obesity, population child obesity surveillance should include measures of WHtR.

## 4. Weight-Related Behaviours

Overweight and obesity are outcomes of energy imbalance and concurrent information on the contextual influences on aetiology are necessary for developing intervention programs. The main modifiable behavioural drivers of overweight/obesity are diet, physical activity, and sleep [24]. Objective measurements collect detailed information of these behaviours and, while desirable, are usually outside the remit of population obesity surveillance surveys. Objective (gold standard) methods to measure diet (e.g., food records and detailed dietary recall), and movement devices for physical activity, sedentary behaviour, and sleep (e.g., accelerometers, pedometers, inclinators, smart phone apps, wearable devices such as FitBits) are associated with high costs, participant burden and researcher expertise and time. Furthermore, the reliability and validity of different movement devices is poorly understood, produce different estimates of these behaviours, and evolutions in device technology may compromise temporal monitoring [46,47].

To minimise cost and participant burden, self-reported measures of indicators of weight-related behaviours, psychometrically tested against gold standards, are often used in population child obesity surveillance. The disadvantages of self-report measures are imprecision associated with respondents’ social desirability and recall bias, and are limited to use among children aged >10 years [48,49]. For children age <10 years, proxy responders (e.g., parents, carers) are a substitute but are also prone to under- or over-reporting because of lack of knowledge of the child’s behaviour (e.g., physical activity at school) [42]. Deciding which measures to use is contingent upon each jurisdiction’s cultural–social–economic factors; however, it is important that measurement tools are valid, reliable, and remain consistent across surveys to monitor trends, and assess the long-term impacts of policies and public health programs that promote healthy weight-related behaviours.

Ideally, child obesity surveillance data should have the capability to calculate outcomes that can be benchmarked against internationally accepted recommendations for weight-related behaviours. In terms of movement, there has been a shift to consolidate guidelines for physical activity, sedentary behaviour, and sleep into one “whole day” (i.e., 24 h) guideline [50,51]. Table 2 lists the recommendations for measurement of behaviours associated with the development of overweight and obesity in children aged 5–17 years.

### 4.1. Dietary Behaviours

Adequate monitoring of food intake and dietary habits is an integral part of any obesity surveillance system. The most pragmatic method of measuring diet in large surveys is using short questions from more substantial food frequency questionnaires that have been shown to capture usual dietary patterns [55]. In most high-income countries, this includes measuring consumption frequency of key “healthy” and “unhealthy” foods, as well measuring certain dietary behaviours or habits that have been associated with weight. At a minimum, these should include frequency of consumption of fruit; vegetables (usually excluding potatoes); sugar-sweetened beverages (including fruit juices); snack foods, usually grouped as those high in salt (potato chips, corn chips, savoury biscuits); and sugar (sweet biscuits, cakes, muffins, donuts), and also “fast foods” or takeaway foods that include pizza, pastries, and hamburgers. Composite measures such as “food risk scores” [56] or “junk food indicies” [57] can be derived from these measures.

It is important to be aware of some of the limitations of short questions for measuring intake. In many cases they can overestimate food intake/energy and need to be validated in the country context. Furthermore, the dynamic changes in the production of food products means measurement tools require regular updating as new foods become available and popular (e.g., energy drinks, ultra-processed foods), and therefore temporal trends can be difficult to track as foods and questions relating to them are constantly evolving.

Another important aspect of diet surveillance systems are a measure of dietary behaviours and habits in the home environment. These can include measures such as breakfast consumption frequency, rules around snacking within the household, and whether food is used as a reward for good behaviour [23].

### 4.2. Physical Activity

Physical activity is an important and modifiable component of energy expenditure that increases from 20% at age one to ~35% at age 18 years [58]. There is convincing evidence that daily physical activity is a protective factor for the prevention (and treatment) of leading noncommunicable diseases, including overweight and obesity [59]. It is estimated that only 20% of children globally participate in the recommended daily prescription of physical activity required for health benefits [60]. Low physical activity levels are associated with lower proficiency in children’s fundamental movement skills and cardiorespiratory and muscular fitness [61].

The most pragmatic measure of children’s physical activity is self-report. The most common measure used in surveillance surveys is a single validated question originally developed by Prochaska and colleagues [62] which asks: Over the past 7 days, on how many days were you/your child engaged in moderate to vigorous physical activity for at least 60 min (this can be accumulated over the entire day, for example, in bouts of 10 min) each day? (Response options 0–7 days).

Ideally, information of children’s proficiency in fundamental movement skills (FMS) should also be collected to inform physical literacy interventions. FMS are the building blocks of physical activity and form the foundation for a range of specific motor skills required in sport and leisure activities. Children who are proficient in FMS are more likely to be physically active, have adequate cardio and muscular fitness, and less likely to have overweight or obesity, compared with children who are not proficient [61]. Additionally, FMS proficient children are more likely to be active and have higher fitness levels in adolescence [63].

FMS can be assessed by a range of objective and subjective tools. There are multiple measurement tools; however, product- and process-orientated assessments with observational methods (direct or videoed) are recommended to determine the prevalence of FMS proficiency [64,65]. The testing time to assess children’s FMS is a limitation that needs consideration, with potentially 30 min per child required. The most common process-orientated measurement tool used globally is the Test of Gross Motor Development–2 (TGMD–2) [66] and product-oriented assessments typically include run time, jump distance, throw, and kick speed.

Similarly, objective field measures of children’s cardiorespiratory and muscular fitness are strongly correlated with physical activity. Low cardiorespiratory fitness during childhood predicts overweight and obesity in adolescence [67] and adulthood [68]. Evidence shows that there has been a substantial decline in children’s cardiorespiratory fitness since 1981, with some stabilisation since 2000 [69]. The most common method for assessing cardiorespiratory fitness in children is the 20-m shuttle-run test [70].

Muscular fitness, or anerobic fitness, is the ability of muscles to undertake short-duration activities and is protective against total and central adiposity and cardiometabolic risk factors [71]. Between 1985 and 2015, there have been substantial declines in children’s muscular fitness assessed by the standing broad jump [72,73]. There is no standard assessment battery for muscular fitness; however, reliable objective field measures include grip strength, sit-up performance, and the standing broad jump [74].

### 4.3. Sedentary Behaviours

Sedentary behaviours are any sitting or reclining activities where energy expenditure is minimal (i.e., ≤1.5 metabolic equivalents, METs) [75]. It is estimated that children spend approximately 60% of their waking hours sedentary [66]. Whether sedentary behaviour in children is a risk factor for health, including overweight and obesity, is contentious [76,77]. This could potentially be because indicators of chronic disease accumulate over time; they are not acute physiological changes. The most common sedentary behaviour in children is screen time (i.e., television, video/computer games) and some reviews indicate there is a small association between children’s television viewing and overweight and obesity [77]. It is hypothesised that this association is confounded by the clustering of other obesogenic behaviours: unhealthy diet, inactivity, and sleep [78]. Children are often the target audience for television advertisements for processed foods and sugar-sweetened drinks [79], and children with high habitual television viewing tend to have diets higher in processed foods and sugar-sweetened drinks, and lower in fruit and vegetables [80]. Although there has been a shift towards more children viewing streaming services (e.g., Netflix), which do not have food advertising, television remains the most popular media device in children [81].

Given the diversity of sedentary behaviours, a single-item question that only assesses television viewing will underestimate overall sedentary behaviour. The quantum shifts in digital technology have resulted in the current generation of children engaging in daily screen time, which can have positive and negative effects on children’s health and well-being. Measuring usage by different digital technologies (e.g., television, smart phone, computer) and purpose (e.g., educational, recreational) can inform intervention strategies. Although children’s screen time is prevalent, it is recommended that domain-specific questionnaires with multiple item domains of sedentariness, such as cultural, educational, transport, and hobbies, are more accurate for assessing total sitting time [82] (e.g., Adolescent Sedentary Activity Questionnaire [83]).

### 4.4. Sleep

More recently, the construct of sleep has been shown to be an important determinant of obesity and overweight in children [84]. The most common way to measure sleep to date has been “sleep duration”, defined as the usual amount of sleep in each 24-h cycle (measured in hours and minutes, combining night-time sleep and day-time naps). A less complicated measure is hours of sleep on an average weeknight as school aged children generally have more regular bed time and wake times on weekdays than during the weekend.

Recommended amounts of sleep differ between age groups and also between countries; for example, the American Academy of Pediatrics recommends between 9 and 12 h per 24 h period for children aged 6 to 12 years and 8 to 10 h for adolescents aged 13–18 years [85], whereas Australian guidelines recommend 9 to 11 h of sleep per night for those aged 5–13 years and 8 to 10 h per night for those aged 14–17 years [50]. Some limited evidence suggests that measurement of “timing of sleep”, such as bed time and wake time may also be important, and a recent systematic review [86] has also drawn attention to other dimensions of sleep, such as sleep quality and sleep efficiency, but there is no internationally accepted definition of these variables.

## 5. Obesogenic Environments

Obesity is now recognised as a complex disease which cannot be solved by influencing individual behaviours alone, but solutions should integrate biological and socioenvironmental determinants such as economics, culture, social networks, and features of the food and built environment, and how these influence behaviours such as eating and physical activity. The increases in the prevalence of obesity globally has been driven primarily by changes in the global food supply, including flooding the market with cheap and highly processed—food that is marketed very effectively, particularly for children [24].

The environmental and commercial determinants of health are increasingly important drivers of obesity, especially in children [24], and measurement of the food and built environments will enhance surveillance surveys that collect information at the individual level. For example, linking child obesity surveillance data by postcode/geolocation with the density of supermarkets, fast food density, and transport data can lead to targeted policy decisions for to address obesogenic factors in local communities.

### 5.1. Home Environment

The home environment and parents are important influences on modifiable risks for child obesity [87]. Parenting styles (i.e., authoritative, authoritarian, permissive, and disengaged) are considered an important intervention point to reduce the risk of obesity in children [88,89]; however, the collection of these data can be beyond the remit of a child obesity surveillance survey. Alternatively, indicators of home and family practices that have been identified as potentially obesogenic for children [90] are feasible within a child obesity surveillance survey. Dietary practices can include the availability of soft drinks in the home, frequency of going to fast food outlets with the family, using sweets as a reward for good behaviour, offering water at meal times, and eating meals in front of the television. Home and family practices that influence children’s movement patterns (i.e., physical activity, screen time, and sleep) include televisions in children’s bedrooms, rules on screen time, driving children to school and other destinations, the availability and access to physical activities, and bed times [91,92,93]. The inclusion of indicators of obesogenic practices can be useful to guide parenting programs, social messaging initiatives, and targeted interventions.

### 5.2. Built Environment

The built environment affects energy balance through opportunities or barriers for physical activity and adherence to dietary recommendations [94]. For example, neighbourhood safety, amenity and design, road safety street connectivity, and traffic volume have been associated with children’s physical activity levels [95]. A better understanding of child obesity, especially in certain subpopulations (e.g., low socioeconomic and culturally and linguistically diverse populations) can be enhanced by triangulating child obesity surveillance data with extant surveys on transport, neighbourhood walkability, and population density, and then applying geographic information system technologies to identify availability of green spaces, sport venues, school location, and school playgrounds.

### 5.3. Food Environment

Measurement of the food environment is complex and for children can include measures such as healthiness of the school food environment, as well as children’s exposure to television and online media, sponsorship of children’s sports, and front-of-pack promotions. The International Network for Food and Obesity/Non-communicable Diseases (NCDs) Research, Monitoring, and Action Support (INFORMAS) has developed a comprehensive tool for benchmarking of national food environments, and has a recommended set of questions for measuring the healthiness of children’s food environment, which has recently been applied in New Zealand [96]. These include measuring the proportion of schools selling sugar-sweetened beverages, using unhealthy food for fundraising, and average number of convenience stores, and density of fast food and takeaway outlets per km^2^ around urban schools. The questions would need to be adapted according to context but would aid in measuring a country or region’s progress towards healthier food environments over time.

## 6. Summary

It is important that government surveillance of child obesity includes measurement of concomitant modifiable behaviours. Determining the prevalence of overweight and obesity alone is insufficient information for policy makers to determine the effectiveness of scaled intervention efforts. The choice of indicators of weight-related behaviours will differ across jurisdictions; however, it is important that the measurement choice remains consistent to estimate temporal trends. While objective measures of eating and movement (i.e., physical activity, sedentary time, and sleep) behaviours are desirable and hold promise for better estimation of these behaviours, the variability in psychometrics and rapid evolution of wearable devices is potentially problematic for examining trends over time and how behaviours may change.

Questionnaires on self-reported behaviours have limitations (e.g., social desirability bias and cross-cultural, age, and sex differences in reporting), but they remain the most feasible and affordable method of collecting population data from, often, thousands of participants. In population surveys, the purpose of self-report is usually to rank and not accurately estimate behaviour. Self-report could potentially be improved through electronic modes of administration of questionnaires by including logic checks and reducing missing data. The advantage of self-report questionnaires is the ability to capture contextual information on domains of behaviour (e.g., physical activity domains of play, sport, active transport; sedentary behaviour domains of reading, homework, screen time) and environmental information (e.g., television in the bedroom, availability of soft drinks in the home) that cannot be captured by wearable devices and other objective measures. In the future, new technology may mean it is more economical and feasible to consider using robust measures, such as logs/diaries or ecological momentary measures of sedentary behaviour and physical activity [97], or assessment instruments such as diet diaries or multiple 24 h recalls for dietary data [98] The integration of technologies into child obesity surveillance surveys, however, need to consider the burden to participants and the impact these technologies may have on determining trends from retrospective data.

The recognition that obesity is a complex and multifactorial issue highlights the need to include some measure of the environmental determinants, such as the food and built environments around where children eat, learn, and play. These measures are particularly relevant in countries with an increasing prevalence of childhood obesity and may lead to changes in government policy. Surveillance systems should be designed to complement national nutrition, physical activity, and general healthy lifestyle policies to assist governments to measure the success of prevention efforts. Effective surveillance systems are essential for preventing and addressing the problem of childhood obesity.

## Figures and Tables

**Table 1 ijerph-17-06812-t001:** Selected examples of population child health surveys.

Survey	Country	Survey Years	Child Ages	Sampling Frame	Anthropometry	Weight-Related Behaviours
Health Behaviour in School-aged Children [25]	Europe, North America (*n* ~ 45–50)	Every 4 years: 1982, 1985–1986, 1993–1994, 1997–1998, 2001–2002, 2005–2006, 2009–2010, 2013–2014, 2017–2018.	11, 13, 15 years	Cross-sectional nationally representative samples (*n* ~ 230,000)	SR (h/w): BMI, OW/OB based on WHO growth reference for age.	Diet: Breakfast, F and V, soft drinks, eat sweets, eat dinner with parent
						PA: number of days over the past week during which they were physically active for a total of at least 60 min.
						SB: (2001), how many hours per day they use a computer for email, internet, or homework in their spare time on weekdays and at weekends. How many hours per day they played games on a computer or a games console in their spare time on weekdays and at weekends.
						Sleep (2005) difficulties in getting to sleep
Childhood Obesity Surveillance Initiative [26]	WHO European Region (*n* = 13)	2007/2008, 2009/2010	6–9 years	Cross-sectional nationally representative samples (*n* ~ 300,000)	Measured (h/w) BMI, OW/OB based on WHO growth reference for age. OW = +1SD, OB = +2SD	Diet: breakfast F&V, savoury snacks, soft drinks with sugar
						PA: school travel, playing actively/vigorously, practicing sports and physical activities with sport clubs or dancing courses
						SB: TV or using electronic devices
						Sleep: bed/wake up on school days
National Health and Nutrition Examination Survey [27]	US	1960s, annually since 1999.	0–19 years *	Cross-sectional nationally representative samples (home and mobile centres) (*n* ~ 5500)	Measured (w/h/wc) BMI: OW ≥ 85th centile, OB ≥ 95th centile	Diet, 2 × 24 h dietary recall (interview)
						PA, # days active for at least 60/min
						SB, playing with a smartphone or computer, watching TV or movies, or playing video games?
						Sleep: Only > 16 years
National Health Surveys [28]	Australia	2007–2008, 2011–2012, 2014–2015, 2017–2018.	5–17 years	Cross-sectional nationally representative random household survey	Measured (w/h/wc)	Diet, PA
National Child Measurement Programme [5]	England	2007–2008, 2008–2009, 2009–2010, 2011–2012	4–5 years and 10–11 years	Cross-sectional. All children aged 4–5 and 10–11 years residing in England	Measured (h/w)	None
Korean National Health and Nutrition Examination Survey [29]	Korea	1998, 2001, 2005, 2007–2009, 2010–2012, 2013–2015	1–17 years	Cross-sectional nationally representative random household survey	Measured (w/h/wc)	Diet, PA, ST, sleep
China Health and Nutrition Survey [30]	China	1989, 1991, 1993, 1997, 2000	6–18 years	Cross-sectional nationally representative random household survey	From 1991, Measured (w/h/wc)	Diet
		2004, 2006, 2009, 2011, 2015				Diet, PA, sleep, ST
Canadian Health Measures Survey [31]	Canada	2007–2009, 2009–2011, 2012–2013, 2012–2015, 2016–2017, 2018–2019	3–18 years	Cross-sectional nationally representative random household survey	Measured (w/h/wc)	PA

SR = self-report; h = height, w = weight, WC = waist circumference; PA = physical activity; ST = screen time; * adult data also collected.

**Table 2 ijerph-17-06812-t002:** International recommendations for weight-related behaviours for children age 5–17 years.

Behaviour	Indicator
Diet [52,53]	Frequency of consumption of—
	Fruit: 2 serves/day
	Vegetables: 5 serves/day
	Breakfast: daily
	Sugar-sweetened beverages
	Common highly processed snack and takeaway foods
**24-Hour Movement Guidelines for Children and Youth (Aged 5–17 years)** [12,43]	
Physical activity [54]	Accumulate at least 60 min of moderate- to vigorous-intensity physical activity daily, >60 min daily will provide additional health benefits.
	Most of daily physical activity should be aerobic. Vigorous-intensity activities should be incorporated, including those that strengthen muscle and bone, at least 3 times per week.
Sedentary behaviour [51]	Limiting sedentary recreational screen time to no more than 2 h per day.
	Breaking up long periods of sitting as often as possible.
Sleep [51]	Uninterrupted 9–11 h of sleep per night for those aged 5–13 years and 8–10 h per night for those aged 14–17 years.
	Consistent bed times and wake times.

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
