# Peer review of "Elements of Effective Population Surveillance Systems for Monitoring Obesity in School Aged Children"

_ijerph, 2020, doi:10.3390/ijerph17186812_

Round 1

Reviewer 1 Report

Paper provides an overview of child obesity surveillance systems globally, and highlights main components and other types of survey data that can enhance our understanding of child obesity. Paper argues that in addition to measures of adiposity, including body mass index, waist circumference, effective surveillance must also include measures of weight-related behaviours including diet, physical activity, sedentary time, and sleep. Overall paper is well written and has focused on an important area of public health concern. I have following comments which can help to improve the paper further. 

Although paper has discussed several areas related to child obesity, however paper lacks in identifying the some specific areas/gaps for future research. Paper may point out some disagreements among scholars in this line of the literature.

As argued by the authors, child obesity is a global problem. Paper may provide some insights on the relationship between economic development of countries and the extent of child obesity.

Is there any research, how does the technology may better help reducing this problem. Though, oppositely, youtube childern videos, video gaming and cartoons are likely to increase this problem. 

Self reported questionnaires have limitations. What are the best alternative ways?

Can paper discuss some specific papers (or at least underlying logic) about the role of built and food environments in this problem.  

Author Response

Reviewer comment Paper provides an overview of child obesity surveillance systems globally, and highlights main components and other types of survey data that can enhance our understanding of child obesity. Paper argues that in addition to measures of adiposity, including body mass index, waist circumference, effective surveillance must also include measures of weight-related behaviours including diet, physical activity, sedentary time, and sleep. Overall paper is well written and has focused on an important area of public health concern. I have following comments which can help to improve the paper further. 

Authors response. Thank you

Reviewer comment Although paper has discussed several areas related to child obesity, however paper lacks in identifying the some specific areas/gaps for future research. Paper may point out some disagreements among scholars in this line of the literature.

As argued by the authors, child obesity is a global problem. Paper may provide some insights on the relationship between economic development of countries and the extent of child obesity.

Authors response. We have added the following text as a brief summary of the regional and socioeconomic disparities in the distribution of children who have obesity.

In 1975 the global prevalence of obesity in children was ~0.8%, increasing to 5.6% in girls and 7.8% in boys in 2016 [1]. There are regional differences with higher prevalences among children from Polynesia, Micronesia, Central Latin America, Middle East and North Africa, and high-income English-speaking countries. The data shows an emerging plateau in the prevalence among children from high-income regions but an increase in high-come Asian countries. There are also socioeconomic differences within jurisdictions; in high-income regions the prevalence is higher among children and in low socioeconomic areas conversely, in low-and middle-income countries the prevalence is higher children from high socioeconomic areas [2].

Reviewer comment Is there any research, how does the technology may better help reducing this problem. Though, oppositely, youtube childern videos, video gaming and cartoons are likely to increase this problem. 

Author’s response Yes, there have been many studies examining the positive and the negative effects of children’s use of screen devices. We agree this is an important area, but feel it is beyond the scope of our paper. We have however added the following text at line 249

The quantum shifts in digital technology mean children now engage in daily screen-time which have positive and negative effects on children’s health and well-being. Measuring usage by different digital technologies (e.g., television, smart phone, computer) and purpose (e.g., educational, recreational) can inform intervention strategies. Although children’s screen-time is prevalent it is recommended that domain-specific questionnaires with multiple items domains of sedentariness such as cultural, educational, transport, hobbies are more accurate to assesses total sitting time [58]. (e.g., Adolescent Sedentary Activity Questionnaire [57])

Reviewer comment Self reported questionnaires have limitations. What are the best alternative ways?

Author’s response We agree that this is important to highlight and have added the following addition to line 311;

“and they remain the most feasible and affordable methods of collecting population data from, often, thousands of participants”.  

In the section following this we add how self-report can be improved and we have also added the following sentence about alternative methods:

In the future new technology may mean it is more economical and feasible to consider using robust measures such as logs/diaries or ecological momentary measures of sedentary behaviour and physical activity [74] or assessment instruments such as diet diaries or multiple 24hrs recalls for dietary data [75] The integration of technologies into child obesity surveillance surveys however need to consider the burden to participants and the impact these technologies may have on determining trends from retrospective data.

Reviewer comment Can paper discuss some specific papers (or at least underlying logic) about the role of built and food environments in this problem.

Author’s response In section 4 of the paper starting at line 278 we outline the contribution of the environmental and commercial determinants of health on obesity and have included the following points in this section.

“Obesity is now recognised as a complex disease which cannot be solved by influencing individual behaviours alone but solutions should integrate biological and socioenvironmental determinants such as economics, culture, social networks and features of the food and built environment and how these influence behaviours such as eating and physical activity . The increases in the prevalence of obesity globally has been driven primarily by changes in the global food supply, including flooding the market with cheap and highly processed, food that is marketed very effectively, particularly for children”.  

Reviewer 2 Report

The authors have provided a well-written review of surveillance systems for childhood obesity. They have also provided a nice argument for the inclusion of behaviors and other factors as part of a surveillance effort.

Overall, I have little to recommend or provide constructive criticism for. A few points are noteworthy, however:

1) Would the authors consider an understanding of how individual behaviors (eg, consumption of highly processed and takeaway foods) intersect with others (eg, consuming such foods during sedentary time in front of at television), or whether or not certain protective behaviors exist, such as healthy family meals at a regular time (without the TV on)? The detrimental or health-promoting values of many of these behaviors has been documented. The question is can we base monitoring systems off of them to help families and to help guide interventions and resource allocation.  Some commentary by the authors would be valuable.

2) Secondly, the distinction between true surveillance systems, which can be costly and logistically difficult to carry out, and monitoring systems which provide a good sense of trends in behaviors, but without the financial or logistical burden of surveillance systems, is probably a good point for discussion. This is because many localities and even nations may be severely challenged to carry out surveillance in the purest sense of the word.

3) Finally, please use People First Language when discussing obesity. People First Language has been espoused by the Obesity Society and many academic and advocacy societies as well as several prominent journals. In People First Language the child, or individual is considered. The individual is not objectified as a disease. So, for example, at Lines 172, 309 and 318 phrases should rather read: : "less likely to have overweight or obesity," "school had overweight or obesity," and "more likely to have overweight or obesity," respectively.

Author Response

Reviewer 2

Reviewer comment The authors have provided a well-written review of surveillance systems for childhood obesity. They have also provided a nice argument for the inclusion of behaviors and other factors as part of a surveillance effort.

Author’s response Thank you.

Reviewer comment Overall, I have little to recommend or provide constructive criticism for. A few points are noteworthy, however:

1) Would the authors consider an understanding of how individual behaviors (eg, consumption of highly processed and takeaway foods) intersect with others (eg, consuming such foods during sedentary time in front of at television), or whether or not certain protective behaviors exist, such as healthy family meals at a regular time (without the TV on)? The detrimental or health-promoting values of many of these behaviors has been documented. The question is can we base monitoring systems off of them to help families and to help guide interventions and resource allocation.  Some commentary by the authors would be valuable.

Author’s response Thank you and we agree this is an important discussion point. Given the depth of understanding the breath of home and family influences on child obesity we have added a summary section titled 4.1 Home environment (line 292).

4.1 Home environment

The home environment and parents are important influences on modifiable risks for child obesity [71]. Parenting styles (i.e., authoritative, authoritarian, permissive, and disengaged) are considered an important intervention point to reduce the risk of obesity in children [72, 73] however the collection of these data can be beyond the remit of a child obesity surveillance survey. Alternatively, indicators of home and family practices that have been identified as potentially obesogenic for children [74] are feasible within a child obesity surveillance survey. Dietary practices can include the availability of soft drinks in the home, frequency of going to fast food outlets with the family, using sweets as a reward for good behaviour and offering water at meal-times and eating meals in front of the television. Home and family practices that influence children’s movement patterns (i.e., physical activity, screen time and sleep) include televisions in children’s bedrooms, rules on screen-time, driving children to school and other destinations , the availability and access to physical activities and bed-times [75-77]. The inclusion of indicators of obesogenic practices in the can be useful to guide parenting programs, social messaging initiatives and targeted interventions.

2) Secondly, the distinction between true surveillance systems, which can be costly and logistically difficult to carry out, and monitoring systems which provide a good sense of trends in behaviors, but without the financial or logistical burden of surveillance systems, is probably a good point for discussion. This is because many localities and even nations may be severely challenged to carry out surveillance in the purest sense of the word.

Author’s response Thank you and we agree this point should be included. We have added a section on considerations for population surveys, and included a section on active and passive surveillance at line 75

Population health surveillance is the on-going systematic collection, analyses, interpretation and, timely dissemination of the data to decision makers responsible for prevention and control [15]. Surveillance data can be used to measure the need for interventions and for directly measuring the effects of population-wide interventions. Active surveillance requires trained field teams to collect the data and while it provides the most accurate and timely information, active surveillance can be expensive. An alternative method to collect population level information on child obesity is through passive surveillance and other health monitoring systems such as medical records. Passive surveillance is less expensive than active surveillance but limited because data are not collected systematically and because the data is collected through a network of systems may have variability in the completeness, quality and timeliness of the data.

Details on population surveillance survey designs and data analyses are beyond the scope of this paper, so we present a general overview of factors to consider. Table 1 shows there are different survey methods to monitor population-level child obesity which include mobile health vans, households, and schools. The methodology will differ across jurisdictions according to a range of factors, including expertise and funding. Surveys such as the US National Health and Nutrition Examination Survey [10] use mobile health vans staffed with specialists’ health assessment collect extensive behavioural and biomarker data but are expensive to conduct. Household and school-based surveys are relatively low-cost but are often unable to collect extensive behavioural and biomarker data.

The sampling frame must replicate the population for the estimates to be generalizable. For example, ensuring the frame covers all demographic characteristics of interest such as age, sex, socioeconomic status, geography (rural/urban) and cultural/racial background. A sample of children are random selected and invited to participate in the survey. Written consent is typically an ethics requirement however the governments in the United Kingdom [5] and the US state Arkansas [16] have legislated data collection with opt‐out consent for their childhood obesity monitoring and screening programs.

The measurement of weight-related behaviours, and how to measure each indicator, will differ across jurisdictions. The survey instruments need to reflect cultural traditions and customs of dietary and physical activity behaviours. One of the purposes of population surveillance is to examine temporal trends so the methodology must remain consistent across survey years. Ideally, data should be captured electronically using computers or hand-held and other mobile devices. In contrast to pen and paper, electronically data entry reduces errors through logic checks, missing data, minimizes cost (e.g., data entry-re-entry) and, ready for cleaning and analyses.

3) Finally, please use People First Language when discussing obesity. People First Language has been espoused by the Obesity Society and many academic and advocacy societies as well as several prominent journals. In People First Language the child, or individual is considered. The individual is not objectified as a disease. So, for example, at Lines 172, 309 and 318 phrases should rather read: : "less likely to have overweight or obesity," "school had overweight or obesity," and "more likely to have overweight or obesity," respectively.

Authors response Thank you, we completely agree that the language to describe obesity should be People First and have corrected the instances where we did not use this, as mentioned in the afore-mentioned lines.

Reviewer 3 Report

This is an article within the scope of Child and Adolescent Health, addressing the theme Childhood Obesity. It is a current and relevant topic in the field of public health, being of international interest. The authors follow the journal template with minimal need for corrections, and it is in conditions to be approved for publication, since the implicit hypothesis contemplates the object of study, its respective construction of the objectives section with the appropriate method (which can be improved) to describe the actions taken in the objective section and the results are derived from the paths taken in the method section. In the discussion, the authors should strive to build the discussion of your results in the first part of the discussion section, leaving the second part for convergence and divergence with the studies available in the PubMed databases and providing more elegance in the third part of the discussion section, leaving explicit the erudition of his field of knowledge. Finally, they should highlight the limitations of this study and the contributions that justify the reasons for this article to be published in fact by this journal.

Author Response

Reviewer 3

Reviewer comment This is an article within the scope of Child and Adolescent Health, addressing the theme Childhood Obesity. It is a current and relevant topic in the field of public health, being of international interest. The authors follow the journal template with minimal need for corrections, and it is in conditions to be approved for publication, since the implicit hypothesis contemplates the object of study, its respective construction of the objectives section with the appropriate method (which can be improved) to describe the actions taken in the objective section and the results are derived from the paths taken in the method section. In the discussion, the authors should strive to build the discussion of your results in the first part of the discussion section, leaving the second part for convergence and divergence with the studies available in the PubMed databases and providing more elegance in the third part of the discussion section, leaving explicit the erudition of his field of knowledge. Finally, they should highlight the limitations of this study and the contributions that justify the reasons for this article to be published in fact by this journal.

Authors response Thank you for this constructive advice.  As this is a commentary/ review article and not a research study we have not included a discussion section but have discussed research gaps convergence and divergence of opinions and controversies throughout the article. In the final section (section 5) we have included a summary of our recommendations in view of current surveillance systems as well as a call to include some measurement of food and built environmental determinants in surveillance systems for childhood obesity.  In line 337 we have added that “effective surveillance systems are essential for preventing and addressing the problem of childhood obesity”. As the reviewer has correctly pointed out our review will fit very well with the special issue of the journal on Child and Adolescent Health, addressing the theme Childhood Obesity as surveillance systems are a critical component for addressing the problem.